# Luminal and Tumor-Associated Gut Microbiome Features Linked to Precancerous Lesions Malignancy Risk: A Compositional Approach

**DOI:** 10.3390/cancers14215207

**Published:** 2022-10-24

**Authors:** Vladimir A. Romanov, Ivan A. Karasev, Natalia S. Klimenko, Stanislav I. Koshechkin, Alexander V. Tyakht, Olga A. Malikhova

**Affiliations:** 1Atlas Biomed Group—Knomx LLC, Tintagel House, 92 Albert Embankment, Lambeth, London SE1 7TY, UK; 2N.N. Blokhin National Medical Research Center of Oncology, Ministry of Health of Russia, 24 Kashirskoe Shosse, 115478 Moscow, Russia; 3Center for Precision Genome Editing and Genetic Technologies for Biomedicine, Institute of Gene Biology Russian Academy of Sciences, 34/5 Vavilova Str., 119334 Moscow, Russia

**Keywords:** colorectal cancer, gut microbiome, precancerous lesions

## Abstract

**Simple Summary:**

Studies of gut microbiome in patients with colorectal cancer (CRC) have shown strong alterations of its community structure. Identification of the early signs of changes may allow us to predict the risk of CRC using microbiome and improve the prognosis for such patients. The aim of our study was to evaluate the microbiome composition of luminal and tumor-associated microbiomes in patients with precancerous lesions and early forms of colon cancer. We found microbiome composition to be associated with many characteristics of the lesions. Among them, the findings related to luminal samples are of particular value due to the low invasiveness of the sample collection. Our results expand understanding of the microbiota involvement in triggering malignancy and contribute to the development of early-stage colon cancer diagnostics.

**Abstract:**

Colorectal cancer is the third most commonly diagnosed cancer worldwide. Human gut microbiome plays important roles in protecting against it, as well as contributing to its onset and progression. Identification of specific bacterial taxa associated with early stages of colorectal cancer may help develop effective microbiome-based diagnostics. For precancerous lesions, links of their characteristics to luminal and tumor-associated microbiome composition are to be elucidated. Paired stool and tumor brush biopsy samples were collected from 50 patients with precancerous lesions and early forms of colon cancer; their microbial communities were profiled using high-throughput 16S rRNA sequencing. We showed that the microbiome differences between stool and biopsy samples can be to a high extent computationally corrected. Compositionality-aware statistical analysis of microbiome composition revealed its associations with the number of lesions, lesion type, location and malignization pathway. A major determinant of precancerous lesions malignancy risk—the number of lesions—was positively associated with the abundance of H_2_S-producing taxa. Our results contribute to the basis for developing early non-invasive colorectal cancer diagnostics via identifying microorganisms likely participating in early stages of cancer pathogenesis.

## 1. Introduction

Colorectal cancer (CRC) is one of the most frequent forms of oncological diseases worldwide, being in second and third place for men and women, respectively [1]. The malignant tumor is commonly preceded by a precancerous lesion—a polyp that can be timely detected during colonoscopy and excised. Starting from the 2000s, screening led to a reduction in incidence rates in developed countries [2,3]. The presence of colonic polyps is quite prevalent: they are detected in more than half of individuals older than 50 years with an average risk of colon cancer. However, not all are equal by their malignization risk: it depends on the lesion size, localization and type, as well as its growth dynamics [4]. Based on the risk estimated after polyp detection, polypectomy is performed, or close monitoring of the detected lesion is recommended.

There are hereditary syndromes leading to colorectal cancer including germline APC mutation (familial adenomatous polyposis) and germline mutations in mismatch repair genes (Lynch syndrome) [5]. However, the majority of CRC cases are sporadic, and in their pathogenesis the environmental factors, especially diet, prevail over genetics [6,7]. Malignization of a hyperplastic lesion can occur one of two ways, depending on the polyp type—classic adenoma-carcinoma sequence or serrated polyp pathway (the former being responsible for 20–25% of sporadic CRC cases [4,8]).

Recently, the role of the gut microbiome in CRC pathogenesis has been extensively studied. The concept of “oncogenic microbiome” finds support in an increasing number of experimental and observational studies, most of which are based on high-throughput sequencing as the main method for microbiome profiling. The following mechanisms of involvement have been established:Genotoxicity of specific bacterial strains—for example, colibactin-producing polyketide synthase (pks)+ *Escherichia coli* and enterotoxigenic *Bacteroides fragilis* [9,10];Modulation of signal pathways—for example, *Fusobacterium nucleatum* adhesin, FadA, binds to E-cadherin and activates oncogenic Wnt pathway [10,11];Metabolism of dietary and other components into DNA-damaging compounds—for example, production of deoxycholic acid from cholic acid (one of bile acids) by gut bacteria [9] and of genotoxic hydrogen sulfide (H_2_S)—by sulfur-metabolizing taxa;Metabolism of dietary and other components to the compounds with anticancer activity, for example, production of butyrate from dietary fibers, which works as a HDAC inhibitor in cancer cells, but not in intestinal epithelial cells (butyrate paradox) [12,13];Maintenance of chronic inflammation known to be one of the driving forces in CRC development [14].

A plethora of associative studies have compared gut microbiome composition between the CRC patients and healthy controls; meta-analyses and reviews suggest that some of the major associations are consistent across different human populations [15,16,17]. Application of machine-learning techniques demonstrated the possibility of highly accurate classification of subjects into CRC-positive/negative ones based on their microbiome. Inclusion of subjects with early stages of cancer along with the more advanced stages provides the opportunity to divide the microorganisms associated with the disease into the “drivers” (linked to earlier stages and putatively causal ones) and “passengers” [18,19]. Interestingly, the taxa found in early precancerous lesions as well as in the CRC at further stages of progression include those that typically occur in the oral cavity [18,20].

Investigation into patients with precancerous lesions is especially important as this may shed light onto the unknown mechanisms of microbiome participation in CRC onset and development, as well as help develop novel diagnostic and therapeutic approaches. Although the disease signature can be more pronounced in the mucosa-associated microbial communities (on the tumor), stool represents a more useful sample type for screening and diagnostics, as it can be collected in a non-invasive manner and contains information about the whole colon. However, the tumor-dwelling opportunistic pathogens can be underrepresented in feces and thus poorly detectable using high-throughput sequencing. These considerations support the need for exploring the relationship between stool and tumor-associated samples via a paired analysis. Noteworthy, few studies have been focused on early-stage lesions with respect to their type and localization [14]. In the present study, we describe fecal and tumor-associated microbiome of the patients with precancerous lesions and early forms of colon cancer, and evaluate the links of microbial community features with miscellaneous lesion characteristics.

## 2. Materials and Methods

### 2.1. Study Design

The cohort included 50 patients with histologically verified epithelial lesions of colon and rectum that included hyperplastic polyps, sessile serrated lesions, adenomas with high- and low-grade dysplasia and high-grade adenocarcinomas (verification was conducted via histological examination of the biopsy material under a microscope) (Table 1). The exclusion criteria were as follows: previous treatment of primary colon lesions; serious accompanying pathology of other body systems and organs precluding the planned examination; antimicrobial therapy within 3 months prior to the study. After the complete clinical examination and morphological confirmation, the patients who signed informed consent were included in the study. For each subject, two samples were collected: a lumen microbiome sample was self-collected by each patient as a stool sample, while a mucosal sample was obtained with a brush biopsy from a lesion. When multiple epithelial lesions were discovered during the examination, a biopsy was performed from the largest of them.

### 2.2. Biosamples Preparation and DNA-Sequencing

The samples were processed in two batches using the following protocol. DNA was extracted from stool and biopsy samples using the Qiagen Power Fecal PRO kit according to the manufacturer’s instructions. Amplification of V4 of the 16S rRNA gene was conducted using the following primers: modification of 515F (5′-GTGBCAGCMGCCGCGGTAA-3′) [21] and Pro-mod-805R (5′-GACTACNVGGGTMTCTAATCC-3′) [22]. The second amplification round was performed using standard Illumina indices with adapters. Both PCR rounds were done using the PCR buffer (Evrogen, Russia) and a Bio-Rad CFX-96 amplifier. PCR products were purified with the DNA Cleanup Mini kit (Evrogen, Russia). DNA concentration was measured with a Qubit fluorometer (Invitrogen, USA) using a Quant-iT dsDNA High-Sensitivity Assay Kit. Purified amplicons were mixed equimolarly according to the obtained concentrations. Further library preparation and sequencing were conducted using MiSeq Reagent Kit v2 (500 cycles) and MiSeq sequencer (Illumina, USA) according to the manufacturer’s recommendations. Primary processing (barcodes extraction) was done as described previously [23]. The post-quality trimmed read pairs were merged with SeqPrep; the resulting average read length was 252 bp.

### 2.3. Data Preprocessing and Primary Analysis

For each PCR batch, a negative control library was prepared and sequenced together with the ordinary samples; from the latter, the reads likely belonging to technical artifacts (contaminants) were removed (Appendix A). Further, the data were analyzed in the Knomics-Biota system [24] (https://biota.knomics.ru/ (accessed on 1 October 2022)): the reads were filtered using the DADA2 algorithm [25] to obtain the ASVs (amplicon sequence variants). Taxonomic classification of ASVs was completed using a classifier implemented in QIIME2 [26,27] and trained on the SILVA v.138 database [28], preprocessed using RESCRIPt (https://github.com/bokulich-lab/RESCRIPt (accessed on 1 October 2022), Quast et al., 2013, Creative Commons Attribution 4.0 License (CC-BY 4.0)). The database 16S rRNA sequences were trimmed according to the primers used and aggregated at 99% similarity threshold.

Relative abundance tables at the levels of species, genus, etc., were obtained by summing the values of their ASVs. Alpha diversity was calculated in two ways—Shannon and chao1 indices—using the species-level tables rarefied to the lowest preprocessed read count per sample (*n* = 2442). Pairwise dissimilarity of samples was evaluated via Aitchison metric at various taxonomic levels (for Aitchison metric calculation, each zero-abundance value was substituted with a 0.5 pseudocount).

### 2.4. Statistical Data Analysis

Due to the small sample size, values for some of the categorical factors from the metadata (lesion characteristics) were aggregated (Table 1). Probable malignization pathway was identified based on the data on mutations, macroscopic characteristics and histology. Overall, it yielded 32 adenoma-carcinoma sequence and 17 serrated polyp pathways cases (for one patient, the data allowed us to determine the pathway that was not provided).

Comparison between the stool and biopsy samples was performed via alpha diversity, beta diversity and microbial balances (the two former were acquired using species abundance table). Beta diversity was analyzed through Aitchison distance using permutational analysis of variance (PERMANOVA) with stratification by the pairs of samples from the same subject; correction for alpha diversity (Shannon index) was completed, as there is a mathematical relationship between the alpha and beta diversity [29]. Alpha diversity was compared using the Wilcoxon rank sum test. Relative abundance of individual taxa in stool and biopsy was performed at the level of species using the nearest balance [30] method according to the following mixed linear model:balance ~ sample_type + (1|patient_id)(1)
where balance is the sought nearest balance; sample_type—type of sample (stool/biopsy); patient_id—identifier of subject; (1|…)—random effect.

To refine the results, the nearest balance was iteratively applied 500 times to 50% of randomly selected samples. The taxa assigned to numerator in >80 iterations (or to denominator in >80 iterations) were considered reproducible and included in the final balance.

Search for associations between the stool/biopsy microbiome composition and measured parameters (Table 1) was conducted via alpha diversity, beta diversity and microbial balances (the two latter were acquired at different taxonomic ranks), as well as the cooperatives of microbial species. Beta diversity (Aitchison distance) was analyzed using PERMANOVA with correction for the batch and alpha diversity. Links of metadata with the alpha diversity were identified using a mixed linear model, where the batch was a random effect. Only for those factors that were significantly linked to general microbiome composition (via beta diversity) was analysis of microbial balances performed (as the nearest balance algorithm does not include a statistical significance evaluation). Reproducibility of taxa in the microbial balances was estimated as described above for the sample type (stool/biopsy) analysis.

Analysis of microbial cooperatives was conducted separately for different sample types and batches and included the following steps: microbial network construction, identification of clusters of co-abundant species, construction of balances from these clusters and application of a linear model to predict the balance fitting each given lesion characteristic. Co-abundance networks of microbial species were obtained using SPIEC-EASI algorithm [31] with the Meinshausen–Bühlmann method for correlations detection (other parameters: number of subsamples—10; number of lambda iterations—10; minimum value of lambda—0.2). Only the species with abundance >10 reads in >10 samples were considered in this analysis. Cooperatives (clusters of co-abundant species) were derived from the resulting network using the Louvain method [32]. For each cluster, a special balance was constructed including all taxa from the cluster as a numerator, and all the remaining taxa—as a denominator.

For comparison with the compositional approach, we additionally performed a component-wise analysis. The taxa proportions in the samples were calculated after rarefaction of abundance tables to an equal number of reads per sample (*n* = 2442). These tables were normalized using the arcsine square root transformation. Each taxon and factor were tested for association using a mixed effect model with correction for alpha diversity as a fixed effect, and for batch—as a random effect.

In all the above-mentioned analyses that included multiple tests, an adjustment for multiple testing was performed using the Benjamini–Hochberg method.

The microbial balances at the levels of species, genera and families significantly associated with any of the lesion characteristics were validated using external data. These were the published WGS datasets on stool samples of healthy controls and patients with different stages of CRC [18]. The sample included:251 healthy controls (Healthy)—subjects with no lesions according to colonoscopy or with up to two small (<5 mm) polyps;patients with more than three adenomas with low-grade dysplasia (MP, *N* = 67);patients with stage 0 CRC (S0, *N* = 73);patients with stage I and II CRC (SI_II, *N* = 111);patients with stage III and IV CRC (SIII_IV, *N* = 74).

Precomputed taxonomic profiles downloaded from the publication were obtained via BLASTn alignment of sequences on All-Species Living Tree Project (LTP) of the SILVA database. The balance values were calculated using the relative abundance values at the level of species. In case of unclassified species from a certain family or genus, we summed up abundances of all species from the respective clade. Then, the balance values were compared between each group of CRC patients (MP, S0, SI–II, SIII–IV) and healthy controls using a linear model. Since this analysis was carried out for the purpose of validation, no correction for multiple testing was conducted therein.

## 3. Results

After preprocessing, the number of reads was 31,449 ± 14,878 per sample (stool: 31,150 ± 17,382; biopsy: 31,748 ± 12,037). Comparison of taxonomic composition between two batches revealed significant differences (PERMANOVA with correction for the Shannon index and sample type, *p* = 0.0002, R^2^ = 4%). Therefore, all further unpaired analyses were conducted with the correction for the batch.

The lumenal and tumor-associated microbiome compositions were clearly different: the intra-individual dissimilarity between the biopsy and stool samples (29.60 ± 3.50, Aitchison distance) was higher than the inter-individual ones stratified by type (stool 22.65 ± 3.62; biopsy 21.2 ± 3.11) (Figure 1A). Overall, microbiome composition on the level of species was significantly associated with the sample type (paired PERMANOVA with correction on Shannon index, *p* = 0.0001, R^2^ = 27%). A more detailed comparison of stool and biopsy using the nearest balance method [30] showed 20 species reproducibly associated with the stool samples, and 14 species—with biopsy (Appendix A). The stool vs. biopsy distinction was emphasized by the absolute results of area under the curve analysis (average AUC on the test set across 500 iterations of cross-validation was equal to 1 ± 0). As seen on the PCoA plot, the clouds of biopsy and stool samples are clearly distinct and the lines connecting paired samples are visually parallel. This indicates that the differences between a stool and a biopsy sample are similar across the individuals. Therefore, these differences can be corrected in an analytical way, at least partially. In this regard, we performed correction using the ilr transformation [33]: the samples were transferred to ilr space, where the stool samples were shifted by the difference between the vectors of mean biopsy composition and mean stool composition. Then, the inverse ilr transformation was applied to obtain the bacterial abundances. As can be seen, the stool samples and biopsies became more similar (Figure 1B,C). The effect size of the differences within them was higher that the differences between the stool samples collected from the same individual within a 2 week period (assessed using the data from [34]) but lower than the inter-individual differences (19.6 ± 2.8, Aitchison distance, Figure 1C).

Alpha diversity of stool samples was higher than that for the biopsy samples—for both Shannon (Wilcoxon rank sum test, *p* = 2 × 10^−6^) and chao1 indices (*p* = 0.0003, Figure 1D–G).

The associations of microbiome composition in general with each lesion characteristic (i.e., via permutational analysis of beta diversity, see Methods) were investigated with the correction for the batch and Shannon index (Figure 2). Significant links with species-level microbiome composition were observed for the lesion size and location for the biopsies (PERMANOVA, FDR = 0.09 and FDR = 0.02, R^2^ = 3% and R^2^ = 5%, respectively). The number of lesions was linked to microbiome: for biopsy samples—on the phylum, class and family levels (FDR < 0.02, R^2^ > 5%); and for the stool—on the phylum, class and order levels (FDR < 0.07, R^2^ > 4%). The number of lesions was negatively correlated with chao1 diversity index in biopsy samples (mixed effect model, FDR = 0.0495, Figure 2).

At the level of individual microbial taxa, the nearest balance revealed the balances of specific taxa responsible for these relations. Firstly, for the lesion location and size, the species-level balances are shown for the biopsies in Figure 3. Noteworthy, there was an overlap of involved taxa between the two parameters: similarly, to the left-sided lesions, those with size >10 mm had higher relative abundance of *Alistipes shahii,* and lower—of *Lachnospiraceae* UCG-010.

As for the number of lesions, its stool and biopsy balances were partly similar. At the phylum-level, both were negatively associated with Verrucomicrobiota (included in the denominator of the two respective microbial balances; Appendix A). As for the positive associations, they were more sample type-unique, and included, among others, Fusobacteriota—for the biopsy—and Desulfobacterota—for the stool. Similar effects were observed at other high taxonomic levels (family to class).

For the three balances detected on the level of species, genus and families (for biopsy samples associations with lesion location, size and number), we conducted an additional validation using the published data on stool microbiome of patients with different CRC stages [18]. We observed a moderate but significant increase in the balance associated with lesion size on 1st stage of CRC compared to healthy controls (*p* = 0.0469). The tendency was also observed for the balance associated with left tumor location to be higher in patients with 3rd and 4th stages of CRC compared to healthy controls (*p* = 0.0990) (Appendix A).

To assess the effect of the compositionality adjustment, we also performed a classic component-wise analysis. The significant associations between taxa and factors partly overlapped with those obtained using balances: the unclassified species *Lachnospiraceae*_GCA-900066575 was associated with the right location and *Sutterella* genus was negatively associated with lesion size (FDR < 0.05) in biopsy samples. There was another association not presented while analyzing the balances—between lesion size and abundance of the *Burkholderiales* order in biopsy samples (FDR < 0.05).

Finally, we evaluated whether any microbial cooperatives (co-abundant groups of taxa) were associated with lesion characteristics, using a compositionality-aware approach (see Methods). The cooperatives were calculated for each sample type and for each batch separately (Appendix A). In the first batch, the “adenoma-carcinoma sequence” pathway was associated with the cooperative, including unclassified species from the *Roseburia* genus and *Lachnospiraceae* NK4A136 group, in biopsy samples (FDR = 0.04). In the second batch, there were no significant associations; the most significant trend also included “adenoma-carcinoma sequence” pathway, but the associated cooperative was different (formed by *Alistipes shahii* and an unclassified species of *Oscillibacter* genus; FDR = 0.10).

## 4. Discussion

A recent review has highlighted five major limitations of existing studies linking gut microbiome and CRC pathogenesis [14]. The list included:study of stool samples alone, without mucosal sampling;focus on patients with cancer rather than the ones with the precancerous lesions;lack of consideration for the lesion type and lesion location;use of amplicon sequencing but not the “shotgun” metagenomics.

As can be seen, our study is free from all of these limitations but the last one. We analyzed stool and biopsy with consideration for the lesion type and location in precancerous patients. As for the last point, “shotgun” (WGS) metagenomics is indeed a more powerful method, providing subspecies-level genomic resolution for the most abundant members of the community as well as insights into their functional potential—in particular, into such clinically relevant modalities as virulence and drug resistance. However, 16S studies are generally more cost-efficient thus allowing us to obtain higher sample sizes and statistical power. The two approaches can complement each other, with 16S used for clinical trials to identify perspective microbial targets, and “shotgun”—to obtain their more accurate representations with high resolution.

Our study is the first one to statistically examine the links of pre-CRC microbiome composition with the lesion characteristics using a compositionality-aware approach (specifically, via our novel method—the nearest balance [30]). Compositional statistical algorithms provide more accurate results by accounting for the fact that the microbiome sequencing data inherently do not provide the absolute quantity of each taxon, but only their relative abundance (i.e., they are compositional). This idea finds fertile ground in the studies linking microbiome with diet and diseases [35,36,37], despite the fact that the community of clinicians and nutritionists are more used to the bacterial percentages, i.e., a component-wise analysis. The output of the nearest balance method is highly interpretable: it is a microbial balance, having one or more taxa in the numerator and one or more taxa in the denominator—denoting two microbial groups that are positively and negatively associated with a certain factor, respectively.

The compositional approach allowed us to identify the stool and tumor microbial balances reproducibly associated with three major lesion characteristics. The number of lesions, an obvious grade of dysplasia, was the most highlighted for both sample types. Due to the relatively low sample size, the findings were at the taxonomic levels presenting low dimensionality—family and higher. Noteworthy, independent of the sample type, the denominator (negative correlate of the lesion number) was represented by a sole lineage—Verrucomicrobiota/Verrucomicrobiae/*Verrucomicrobiales*. While multiple representatives are known, one of its most studied member species is *Akkermansia muciniphila*, a beneficial mucosa-dwelling microorganism conferring positive effects on host metabolism. As for the phyla directly associated with the number of lesions, we identified the Desulfobacterota/Desulfovibrionia/*Desulfovibrionales*/*Desulfovibrionaceae* at the intersection between lumen and tumor microbiomes. These related microbial clades include sulfide-producing taxa previously shown to be enriched in early stages of colorectal cancer [18]. In the same study, multiple polypoid adenomas with low-grade dysplasia were associated with the relative abundance of sulfide synthesis pathways. Other studies showed that concentrations of the hydrogen sulfide (H_2_S) produced by these taxa were elevated in the stool samples of CRC patients compared to the healthy controls [38,39]. The H_2_S is known for its genotoxic properties [40,41] and ability to impair colonocyte uptake of butyrate, an anti-inflammatory short-chain fatty acid [39]. For tumor-associated microbiome samples, the number of lesions were also positively associated with the Fusobacteriota/Fusobacteria/*Fusobacteriaceae* abundance. The *Fusobacterium nucleatum* belonging to these clades is one of the taxa most frequently associated with CRC [11]. Many mechanisms have already been shown to explain this correlation. Noteworthy, one of them is the capability to produce H_2_S already mentioned above. Thus, we observe the functional commonality of distinct taxa—the Desulfobacterota that are predominantly in the stool samples, and the Fusobacteriota in the biopsy.

The other two lesion characteristics associated with microbiome were location (reduced to two values—right- or left-sided) and size (reduced to two values—more or less than 10 mm). The associations were detected only for the biopsy samples. Some of the associations intersected between the two parameters: left-sided lesions (more distal) >10 mm in size were characterized by a higher abundance of *Alistipes shahii*, and lower—of unclassified bacteria from *Lachnospiracerae* UCG010. Precancerous lesions with such properties are characterized by higher malignization and recurrence after colonoscopy [4,42]. The balances detected for these two factors were also increased in CRC patients compared with healthy controls, according to our validation on external data. However, the effect size was quite small. The balance associated with the number of polyps was not significantly different between the patients and healthy controls. The reason for this may be that the balances we identified are not features of the disease, but rather indicators of the disease risk. It is known that different stages of CRC are associated with distinct microbial signatures [18,19]. In a similar way, the taxa associated with the characteristics of precancerous lesions may differ from those associated with the very earliest stages of the disease. However, the fact that differences were still observed for the lesions’ localization and size makes the balances associated with these factors more reliable.

Stool and biopsy samples differed quite strongly in their microbiome content; yet, the main differences were consistent between different samples and therefore appropriate for analytical correction. This fact, along with the associations between stool microbiome content and lesions parameters, confirms the predictive value of stool samples for assessing the risk of lesions malignization.

Although we managed to address the inter-batch variability statistically, there were differences between the two batches in one analysis where the correction was not possible—linking microbial cooperatives and lesion type. This suggests that larger sample sizes will help validate our findings. Still, our project is an important step for prioritizing the potential biomarkers and optimizing study design.

## 5. Conclusions

Our study shows that microbiome analysis may be a promising method for establishing the risk of precancerous colon lesions malignization. The tumor and lumenal microbiome profile was associated with the lesion characteristics reflecting this risk. At the same time, the predictive potential of lumenal microbiome, which can be easily estimated from stool samples, was comparable to that of a tumor-associated microbiome. The identified microbiome features included sulfate-reducing bacteria, possibly indicating microbiome involvement in the onset and progression of colorectal cancer. These findings form a support for further extended studies involving a control group.

## Figures and Tables

**Figure 1 cancers-14-05207-f001:**
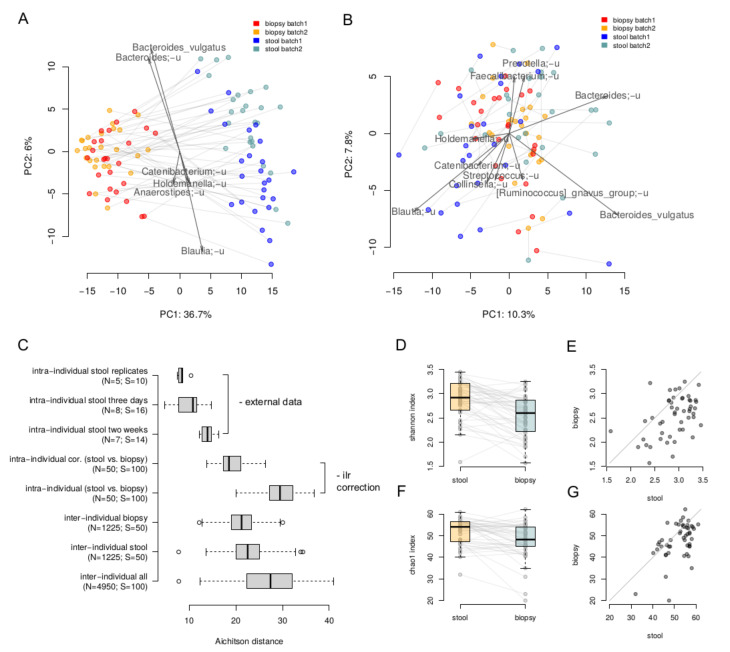
Comparison of paired luminal and tumor-associated microbiomes. (**A**,**B**)—The stool and biopsy compositions are visualized using PCoA (Principal Coordinates Analysis) based on the Aitchison distance. Arrows show the top taxa in terms of the explained variance in given axes. An arrow length is proportional to the percent of variance explained by the taxon. An arrow angle reflects the distribution of this variance between the axes ((**A**)—before the correction; (**B**)—after correction in the ilr coordinates). (**C**)—Intra and inter-individual variation between samples. Variation between stool samples from the same person obtained from the same sample (“intra-individual stool replicates”), with 3 days (“intra-individual stool three days”) and 2 weeks (“intra-individual stool two weeks”) gap was estimated using the previously published data [34]. Variation between stool and biopsy samples of one patient was calculated before (“intra-individual”) and after (“intra-individual cor.”) the correction. The number of distances computed in each group as well as number of samples are given in brackets. (**D**,**E**)—Comparison of alpha diversity of the stool and biopsy samples (Shannon metric). (**F**,**G**)—Comparison of alpha diversity of the stool and biopsy samples (Chao1 metric).

**Figure 2 cancers-14-05207-f002:**
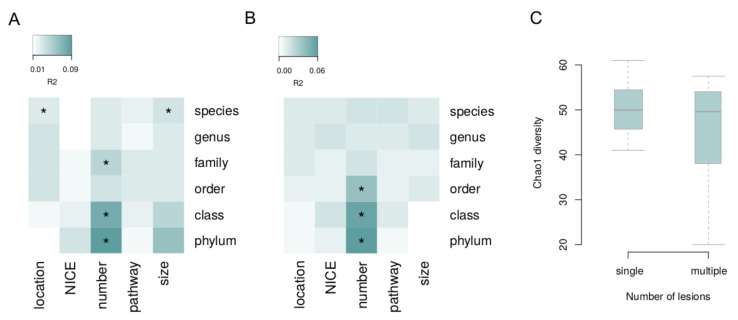
Associations of overall microbiome composition with lesion characteristics. Beta diversity was analyzed at different taxonomic levels. (**A**) Tumor biopsy. (**B**) Stool. (**C**) Biopsy: alpha diversity is linked to the number of lesions. Asterisks denote significant associations (FDR < 0.05).

**Figure 3 cancers-14-05207-f003:**
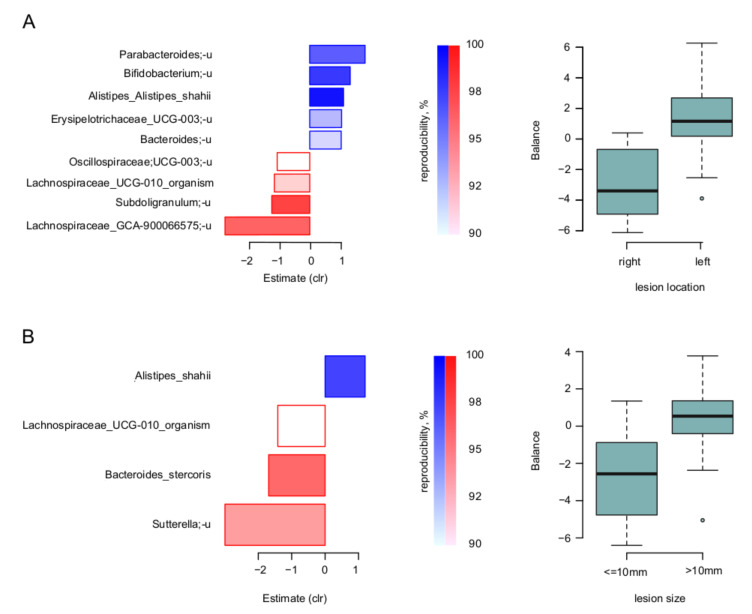
Link of lesion parameters to tumor-associated microbiome: results of “nearest balance” method for the biopsy samples. On the left: bacterial species—reproducible members of the nearest balances revealed in cross validation. The upper blueish bars show the taxa in the numerator in the order of decreasing linear regression coefficient between clr-transformed microbial abundance and factor (*x* axis). The lower reddish bars show denominator taxa. The color tint is proportional to the reproducibility of the taxa in cross-validation analysis (the proportion of iterations in which the taxon was included in the nearest balance). On the right: distribution of the balance values for each parameter value. (**A**) Location of lesion. (**B**) Size of lesion.

**Table 1 cancers-14-05207-t001:** Metadata: factors, aggregation of their values, and treatment in the data analysis. Some of the factors were available for a subset of patients only.

Parameter	Categories before Aggregation (Number of Patients with This Value)	Categories after Aggregation (Number of Patients)	Use in the Analysis
Lesion location	Cecum (3) Ascending colon (10) Transverse colon (4)	Right (17)	Statistical analysis
	Descending colon (5) Sigmoid colon (21) Rectum (7)	Left (33)	
NICE category	1 (10)	1 (10)	
	2 (36) 3 (4)	>1 (40)	
Number of lesions	1 (34) 2 (14) 3(1) 5 (1)	1 (34) >1 (16)	
Size of lesions	<5 mm (11) 6–9 mm (16)	<10 mm (27)	
	10–14 mm (10) 15–19 mm (7) >20 mm (6)	≥10 mm (23)	
Macroscopic characteristic	0–1 s (33) 0–1 p (4) 0–1 sp (13)		For malignization pathway identification only
BRAF mutation	Yes (10) No (15)		
NRAS mutation	Yes (3) No (22)		
Histology	Hyperplastic polyp (9) Sessile serrated lesion (SSL) with low grade dysplasia (13) Adenoma with low grade dysplasia (3) Adenoma with high grade dysplasia (20) Highly differentiated adenocarcinoma (4)		

## Data Availability

The sequencing data are available under accession number PRJNA869306.

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
