# Peer review of "Luminal and Tumor-Associated Gut Microbiome Features Linked to Precancerous Lesions Malignancy Risk: A Compositional Approach"

_cancers, 2022, doi:10.3390/cancers14215207_

Round 1

Reviewer 1 Report

In this manuscript, Romanov et al. uses a human cohort to analyze the luminal and tumor-proximal samples for their bacterial composition in relationship to precancerous lesions. The concept is interesting and hopeful, that the microbiome composition may serve as a biomarker of CRC progression. Unfortunately, it is unclear as to whether their findings make progress towards this goal. The work is heavily focused on analysis of their data set but there is little apparent progress towards the goal that they outline. Crucially missing are healthy controls and the longitudinal follow up as to whether their findings had correlations to extent of disease progression, i.e., the malignancy they propose.

Major concerns:

-       The writing is heavily technical and does not describe the rationale or significance of why different modes of analysis are used. It reads like the authors aimed to thoroughly process their data set. How do the findings of these analyses support their aim of identifying microbiome-based markers of tumor malignancy? What is the physiological relevance of the bacteria identified in their study?

-       In Figure 3, what does “reproducibility” mean? Also the color scale of the reproducibility legend doesn’t match the colors in the bars of the data figure. Most of the bars of darker intensity than is reached in the legend.

-       The same is for Figure 2, the colors in the dataset are darker than what is reached in the legend. What do the asterisks mean, they are undefined.

-       There are no healthy controls examined in this study. How can the authors make appropriate conclusions on the use of microbiome markers for disease when they do not know if these markers are also present in healthy individuals?

Minor concerns:

-       In line 295, the authors write that the necessity of shotgun metagenomics is arguable because it is less cost-efficient than 16S rRNA sequencing. This is quite concerning because there is tremendous scientific value in a whole-genome sequencing approach that actually collects all sequences of microbes rather than sequencing a tiny fraction of the bacterial genome with 16S rRNA sequencing. Furthermore, it is difficult to identify species using 16S-based approaches that target only the V4 region, as done by the authors (Johnson et al. Nat Commun. 2019; PMID: 31695033)

-       Supplementary figures have been omitted. Line 227 refers to supplementary figures 1D-G but this figure only contains 1A-C.

Author Response

In this manuscript, Romanov et al. uses a human cohort to analyze the luminal and tumor-proximal samples for their bacterial composition in relationship to precancerous lesions. The concept is interesting and hopeful, that the microbiome composition may serve as a biomarker of CRC progression. Unfortunately, it is unclear as to whether their findings make progress towards this goal. The work is heavily focused on analysis of their data set but there is little apparent progress towards the goal that they outline. Crucially missing are healthy controls and the longitudinal follow up as to whether their findings had correlations to extent of disease progression, i.e., the malignancy they propose.

We appreciate your constructively critical approach to our manuscript. Please consider our comments to your remarks below.

Major concerns:

-       The writing is heavily technical and does not describe the rationale or significance of why different modes of analysis are used. It reads like the authors aimed to thoroughly process their data set. How do the findings of these analyses support their aim of identifying microbiome-based markers of tumor malignancy? What is the physiological relevance of the bacteria identified in their study?

The compositional-aware approaches are quite novel compared to the component-based approaches used in the field of clinical quantitative microbiome NGS surveys, and they may appear somewhat less intuitive than the latter. With understanding of this nuisance and basing on our prior expertise as microbiome data analysis, we set out to apply the compositional approach to explore the gut microbiome in pre-CRC patients - as this has never been done before, as far as we the authors are concerned. Of note, the approach is represented by a specific method - the “nearest balance” - we have recently developed and for which its advantages have been demonstrated (Odintsova et al, mSystems, 2022). So, the novelty of the approach overall as the specific method in particular, could have made our manuscript look somewhat technical, we agree.

Talking about the microbiome-based biomarkers. Many studies do compare microbiome composition between healthy subjects and patients with a certain established disease. However, in such cases it’s not quite clear whether the observed differences are the result of disease effect, or rather it’s them that contributed to the manifestation of the disease, or both. Therefore, it appears reasonable to compare the lesion characteristics with microbiome features in the patients who might develop CRC in the future. This could also provide hints about whether one should perform colonoscopy having microbiome composition available; and, further on, if it’s justified to remove the polyps. With this in mind, we have modified the abstract to make it clear that the bacterial associations we identified are not the markers per se rather the parameters that can be used to assess the risk of the CRC in the future. The species involved can be viewed as priority candidates to be investigated in a targeted way in future studies.

Considering the physiological relevance of the findings. Firstly, we have identified an interesting link of H2S-producing bacteria with the lesion characteristics that we have summarized in one of our suggestions. Secondly, the gut microbiome represents a complex community with hundreds of members that are involved in multi-layered trophic interactions and many of which can hardly be cultivated without applying advanced techniques, so it’s expectable that not all associative findings in the NGS microbiome surveys can be easily interpreted, especially when it comes to their co-interactions with the host. Particularly, the evidence shows that in the context of disease, for most microbial species one cannot demark certain species as being “good” or “bad” - so are many of the members of the balances we have identified.

-       In Figure 3, what does “reproducibility” mean? Also the color scale of the reproducibility legend doesn’t match the colors in the bars of the data figure. Most of the bars of darker intensity than is reached in the legend.

Thank you for making note of this issue. We have now added the reproducibility description to the figure legend and expanded the interval of heatmap legend. We also changed the levels order in the “location” factor, so that the left-sided lesions (with higher malignization risk) will be treated as “1” in the linear model, and right-sided ones - as “0”. This was done to unify the factors in a way that their higher levels are associated with less favorable prognosis.

-       The same is for Figure 2, the colors in the dataset are darker than what is reached in the legend. What do the asterisks mean, they are undefined.

Indeed, there was a mistake in color legend. The description has been added and the color legend corrected.

-       There are no healthy controls examined in this study. How can the authors make appropriate conclusions on the use of microbiome markers for disease when they do not know if these markers are also present in healthy individuals?

As mentioned above in our answer to your first comment, the discovered microbiome features do not represent the markers of CRC in a strict sense, but rather should be considered as potential contributors to the CRC risk in the future. However, we agree that it is very interesting to look at the values of the identified balances both in healthy controls and in patients with different stages of CRC. For this purpose, we calculated them and investigated their distributions over a large published dataset for the cohort of patients with different CRC stages and healthy controls (https://www.nature.com/articles/s41591-019-0458-7). Please see our amendments to the Methods,  Results and Discussion and new Supplementary figure 3.

Minor concerns:

-       In line 295, the authors write that the necessity of shotgun metagenomics is arguable because it is less cost-efficient than 16S rRNA sequencing. This is quite concerning because there is tremendous scientific value in a whole-genome sequencing approach that actually collects all sequences of microbes rather than sequencing a tiny fraction of the bacterial genome with 16S rRNA sequencing. Furthermore, it is difficult to identify species using 16S-based approaches that target only the V4 region, as done by the authors (Johnson et al. Nat Commun. 2019; PMID: 31695033)

We agree that WGS provides insight into the functional potential of individual members of the microbial community, in particular, such clinically relevant modalities as virulence and drug resistance. Still, it’s yet to become affordable to many research groups and clinical centers. Following your suggestions, we have reformulated this paragraph.

-       Supplementary figures have been omitted. Line 227 refers to supplementary figures 1D-G but this figure only contains 1A-C.

Thank you very much. We’ve corrected this mistake (the reference should be to Figure 1, not to Supplementary Figure 1).

Reviewer 2 Report

The paper is very well written and conceptualized. The topic of interest is also very "in" and relevant. I feel the bioinformatics methods have been explained well and in good detail. A few things I would like to mention:-

1. I did not see the mention of the 16S data being uploaded to a public repository

2. I feel the authors could have restricted themselves to doing a genus level analysis. Species level analysis at 16S level is still very fragile. A lot of unclassified taxa raise some unnecessary questions when the study design is actually pretty strong.

Author Response

The paper is very well written and conceptualized. The topic of interest is also very "in" and relevant. I feel the bioinformatics methods have been explained well and in good detail. A few things I would like to mention:-

Thank you for your appreciation of our work.

  1. I did not see the mention of the 16S data being uploaded to a public repository

Please see the section “Data availability statement”.

  1. I feel the authors could have restricted themselves to doing a genus level analysis. Species level analysis at 16S level is still very fragile. A lot of unclassified taxa raise some unnecessary questions when the study design is actually pretty strong.

We agree that analyzing species with 16S rRNA gene sequencing results in a lot of unclassified representatives.  However, we would like to leave the findings at the species level, since the associations with the lesion’s size and location were detected at the level of species only. Despite the presence of unclassified representatives of genera and families in the balances, there are also taxa identified at the species level with 16S V4 region. In addition, during the review process, we saw that the species balance associated with the polyps size was significantly increased at one of the CRC stages compared with healthy controls when validating our results on a large published WGS dataset. Please see corrections in Methods, Results and Discussion and new Supplementary figure 3.

Reviewer 3 Report

Romanov et al. analyzed, with an innovative compositionally approach, the relationship between stool and mucosal microbiota and the characteristics of colonic pre-cancerous lesions. The study is surely novel and interesting, I have only few comments for the authors:

-       I would highlight in the abstract the comparison between stool and mucosal sample, because one of the main messages of the paper is that stool samples are adequate predictors of tumor associated microbiome in this clinical setting.

-       Lines 56-58 please rephrase the sentence, because in the current form it seems that diet prevails over genetics also in Lynch syndrome and familial adenomatous polyposis. 

-       Please define precisely the type of histological lesions that the authors considered “verified epithelial lesions of colon”, and that therefore could be included in the study.

-       Line 214 and following I would suggest to better explain this important point in order for non-biological readers to better grasp the concept.

-       Beta diversity is a measure of the similarity between two communities. In line 243, what do the authors mean with “ The associations of microbiome composition in general (beta diversity) with each lesion characteristic”? I believe that you found association between taxonomic composition and lesion characteristics, or do you perform other analyses? Please better clarify this point

-       It would be interesting to provide the data regarding the association between lesion characteristics and microbiome using a traditional component-wise analysis, in order to highlight the differences from the novel approach that the authors performed. 

-       Could the authors provide some data on the link between evolution of lesions to CRC and microbiome composition? This would be a great addition to the paper, despite the clearly low statistical power of the association in such a small cohort.

-       Line 317 also other members of the Verrucomicrobiota/ Verrucomicrobiae/ Verrucomicrobiales lineage could have a protective effect, not necessarily Akkermansia. I suggest the authors rephrasing this sentence.

Author Response

Romanov et al. analyzed, with an innovative compositionally approach, the relationship between stool and mucosal microbiota and the characteristics of colonic pre-cancerous lesions. The study is surely novel and interesting, I have only few comments for the authors:

-       I would highlight in the abstract the comparison between stool and mucosal sample, because one of the main messages of the paper is that stool samples are adequate predictors of tumor associated microbiome in this clinical setting.

Thank you for mentioning this. We have added the results concerning stool and biopsy comparison to the abstract.

-       Lines 56-58 please rephrase the sentence, because in the current form it seems that diet prevails over genetics also in Lynch syndrome and familial adenomatous polyposis. 

 We have rephrased the sentence.

-       Please define precisely the type of histological lesions that the authors considered “verified epithelial lesions of colon”, and that therefore could be included in the study.

We have added the clarification in the text.

-       Line 214 and following I would suggest to better explain this important point in order for non-biological readers to better grasp the concept.

We have described this point in more detail.

-       Beta diversity is a measure of the similarity between two communities. In line 243, what do the authors mean with “ The associations of microbiome composition in general (beta diversity) with each lesion characteristic”? I believe that you found association between taxonomic composition and lesion characteristics, or do you perform other analyses? Please better clarify this point

Yes, you are right. Using beta diversity values we are able to explore taxonomic composition in general. Not on the level of specific taxa but rather using information about distances between each sample pair. Statistical tests based on distances (like PERMANOVA, dbRDA) are nonparametric analogs of MANOVA. They can answer the question “is the microbiome associated with the factor” without specifying which specific taxa is responsible for the association. We are interested in these results because now there are no best practices on how to assess p values using compositional-aware methods. The number of balances that can be constructed from the taxonomic tables is so high that you can obtain significant balance for almost any factor. Therefore there are a number of methods (Nearest balance, selbal) which can give you the estimate of best balance for the factor but they cannot answer the question if the association is significant. That’s why we have searched for balances only for the factors which appeared to be significant according to beta diversity analysis (PERMANOVA).

To clarify this points to the readers we have rephrase the sentence.

-       It would be interesting to provide the data regarding the association between lesion characteristics and microbiome using a traditional component-wise analysis, in order to highlight the differences from the novel approach that the authors performed. 

Yes, we agree that it is interesting. We added this analysis, please see corrections in Methods and Results.

-       Could the authors provide some data on the link between evolution of lesions to CRC and microbiome composition? This would be a great addition to the paper, despite the clearly low statistical power of the association in such a small cohort.

Thank you for this suggestion. Following it as well as Reviewer #1 comment we explored the found balances distribution in the large cohort of patients with different CRC stages and healthy controls (https://www.nature.com/articles/s41591-019-0458-7). Please see corrections in Methods, Results, Discussion and new Supplementary figure 3.

-       Line 317 also other members of the Verrucomicrobiota/ Verrucomicrobiae/ Verrucomicrobiales lineage could have a protective effect, not necessarily Akkermansia. I suggest the authors rephrasing this sentence.

Thank you for mentioning this. We have rephrased the sentence.

Round 2

Reviewer 1 Report

The authors have satisfied my concerns.